# On the Need to Distinguish between Insulin-Normal and Insulin-Resistant Patients in Testosterone Therapy

**DOI:** 10.3390/ijms232112730

**Published:** 2022-10-22

**Authors:** Lello Zolla

**Affiliations:** University of Tuscia, 01100 Viterbo, Italy; zolla@unitus.it; Tel.: +39-0761-357100

**Keywords:** insulin resistance, testosterone therapy, hypogonadism, ketone bodies, metabolisms, ketosis, lactate

## Abstract

Male hypogonadism is a disorder characterized by low levels of the hormone testosterone and patients may also have insulin sensitivity (IS) or insulin resistance (IR), such that they show different clinical complications and different metabolic pathways. In this review, we compare metabonomic differences observed between these two groups before and after testosterone therapy (TRT) in order to obtain information on whether the two hormones testosterone and insulin are synergistic or antagonistic. IS hypogonadism uses glucose as the main biofuel, while IR activates gluconeogenesis by the degradation of branched-chain amino acids. The Krebs (TCA) cycle is active in IS but connected with glutaminolysis, while in IR the TCA cycle stops at citrate, which is used for lipogenesis. In both cases, the utilization of fatty acids for energy (β-oxidation) is hampered by lower amounts of acetylcarnitine, although it is favored by the absence of insulin in IR. Increased free fatty acids (FFAs) are free in the blood in IS, while they are partially incorporated in triglycerides in IR. Thus, upon TRT, the utilization of glucose is increased more in IS than in IR, revealing that in IR there is a switch from preferential glucose oxidation to lipid oxidation. However, in both cases, a high production of lactate and acetyl-CoA is the final result, with these levels being much higher in IR. Lactate is used in IS in the glucose–lactate cycle between the liver and muscle to produce energy, while in IR lactate and acetyl-CoA are biotransformed into ketone bodies, resulting in ketonuria. In conclusion, the restoration of testosterone values in hypogonadism gives better results in IS than in IR patients: in IS, TRT restores most of the metabolic pathways, while in IR TRT impairs insulin, and when insulin is inactive TRT activates an ancestral molecular mechanism to produce energy. This evidence supports the hypothesis that, over time, hypogonadism switches from IS to IR, and in the latter case most of the insulin-related metabolisms are not reactivated, at least within 60 days of TRT. However, testosterone therapy in both IS and IR might be of benefit given supplementation with metabolites that are not completely restored upon TRT, in order to help restore physiological metabolisms. This review underlines the importance of using a systems biology approach to shed light on the molecular mechanisms of related biochemical pathways involving insulin and testosterone.

## 1. Introduction

Male hypogonadism is a disorder characterized by low levels of the hormone testosterone [1]. Two main types of hypogonadism have been documented: primary hypogonadism, due to testicular defects, and secondary hypogonadism, where the defect could be in the hypothalamus or in the pituitary gland. It is known that serum testosterone levels decrease after 50 years of age and that at 70 years of age the production rate of testosterone decreases to less than 50% of that of a younger male [1,2]. Hypogonadism affects 6–12% of men aged between 40 and 69 years. Clinically, testosterone deficiency is associated with reduced insulin sensitivity, impaired glucose tolerance, increased fat mass, and elevated triglyceride (TG) and total cholesterol but low HDL cholesterol levels. Hypogonadism is strongly associated with metabolic disorders, including obesity, hypertension, diabetes and dyslipidemia, which are significantly more frequent in hypogonadal male than in eugonadal subjects [2,3,4]. Other symptoms associated with testosterone deficiency include sexual dysfunction, decreased motivation, depressed mood and fatigue [5,6], indicating that testosterone has an influence on the maintenance of bone and muscle mass. Testosterone also exerts a wide range of beneficial physiological effects on body-fat composition, playing a significant role in glucose homoeostasis and lipid metabolism [7]. However, recent evidence suggests that the metabolic alterations revealed in hypogonadism occur through different mechanisms [8,9,10,11], depending on whether patients have high or low insulin levels (regardless of diabetes) [8,12]. For this reason, hypogonadal patients have been classified into insulin-resistant (IR) and insulin-sensitive (IS) categories according to their HOMAi (Homoeostatic Model Assessment for Insulin Resistance index) measures, insulin resistance being defined as the impairment of insulin-mediated glucose disposal by the body [12,13]. This division has induced researchers to gain a better understanding of the biochemical and clinical differences between the two subgroups. Recent studies [13] have shown a positive correlation between testosterone levels and insulin sensitivity. Clearly, in these two different groups the inflammatory mediators increase differently and interfere with insulin signaling in different ways. Low testosterone levels were found to induce insulin resistance (IR) [13,14]. An accumulation of fat can lead to insulin resistance and subsequently to diabetes [15,16,17,18,19,20,21]. In support of this, at low testosterone concentrations, reduced expression of genes associated with glucose transport was observed, including insulin receptor substrate 1 (IRS-1), insulin receptor beta subunit (IR-β), glucose-transporter-type 4 (GLUT4) or AKT serine/threonine kinase-2 (AKT2) and solute carrier family 2 member 4 (SLC2A4) [13,22]. Thus, nowadays it is commonly accepted that young patients with hypogonadism may initially present insulin-sensitivity (IS) states but that, over time, their blood insulin concentrations may increase, leading to insulin resistance (IR) [23] and type 2 diabetes [24].

Cardiac, striated muscular and adipose tissues are insulin-dependent, having higher levels of glucose transporter 4 (GLUT-4). Thus, testosterone treatment of hypogonadal men improved insulin signal transduction [25], since the GLUT-4 receptor is activated in the presence of insulin but modulated by testosterone [26]. In patients with hypogonadism, administration of testosterone in gel formulation for long time periods is effective, with monitoring of effects every 3 to 6 months in the first year and at yearly intervals thereafter [27]. However, the use of TRT is still controversial, showing some benefits, such as better sexual function, better bone mineral density and increased strength, but also increased cardiovascular risk as a result of heart disease.

Currently, little is known about the effect of testosterone on the metabolic and lipidomic pathways, which knowledge could help clinicians to evaluate doses and treatment times. In this regard, a metabonomic comparison of metabolites present in the blood of hypogonadal patients before and after TRT could reveal how tissues respond to testosterone recovery. This might help us better understand the synergy or antagonism between the two hormones and how an endocrinologist should intervene. Recently, an analysis of metabolites present in plasma from two different patient groups, IS and IR, showed different metabolic pathways [28,29]. Through high-resolution mass spectrometry (HRMS) methods [30], a huge number of spectral features in human plasma were revealed and compared [28,29]. It is of note that, since testosterone has a complex and unique regulatory influence on the metabolism of the major tissues involved in insulin action (including liver, adipose and muscle tissues), the metabolomic analysis of plasma, the final collector of all tissues, allowed a holistic investigation, highlighting the importance of a systems biology approach. Finally, since synergic and/or antagonistic interactions between testosterone and insulin exist, by comparing plasma metabolisms recorded in IS and IR hypogonadism patients before testosterone restoration (where testosterone is low in both sets of patients and insulin function in the one case is normal while the other is characterized by insulin resistance) and after testosterone restoration (where testosterone is normal in both but insulin functions are different), interesting information can be obtained on the contributions of the two hormones, both separately and together. Thus, in this review, by comparing metabonomic analyses of IS and IR patients, a comprehensive, simultaneous and systematic profiling of many metabolite concentrations and their fluctuations in response to testosterone therapy will be discussed.

## 2. Metabolomic Comparison between IS and IR Hypogonadism

Figure 1 summarizes and compares the main metabolic pathways recorded in the plasma of IS and IR hypogonadism patients before TRT. The arrows indicate increased or decreased metabolisms in the two hypogonadal subgroups, as well as alterations in intensity. In the absence of an arrow, the metabolism represents a control.

### 2.1. Carbohydrate Metabolism

In IS patients, as expected given the presence of insulin activity, glucose was used in muscle, adipose and liver as the main biofuel. However, imbalances in several other pathways were found, such as the pentose phosphate pathway (PPP), the glycerol shuttle, the malate shuttle, the TCA cycle and lipid metabolism. On the contrary, in the case of IR, glucose metabolism was strongly reduced, and, in the liver, gluconeogenesis was activated, fueled by the conversion of amino acids into glycolytic precursors, branched-chain amino acids (BCAAs) in particular. This explains the individuals with lean body mass and increased fat mass. This different behavior was expected, since GLUT4 expression is reduced in both muscle and adipose tissue in IR hypogonadism [31,32]. As a result, glucose accumulates in plasma, reaching values 2.5 times higher than control values and with respect to IS hypogonadisms [33]. On the contrary, in the liver tissue, where glucose uptake may occur via the GLUT2 transporter, the expression of which is modulated by testosterone as well as glycogen phosphorylase activity [34,35,36], glucose is produced through gluconeogenesis, in agreement with Martin et al. [37].

PPP was strongly upregulated in IS but not in IR, indicating oxidative stress in IS, as also confirmed by increased oxidized glutathione (GS-SG) accumulation, in agreement with Haymana et al. [38].

Interestingly, IS hypogonadism is associated with significant lactate production, which is probably related to the correlation between lactate and testosterone production in rat Leydig cells [14,39]. Surprisingly, in IR, lactate levels were abnormally low [29], because in these patients lactate enters the liver to fuel gluconeogenesis. In both IS and IR, levels of acetyl-CoA were slightly reduced (in IR by 80%). As a consequence, reduced amounts of acetyl-CoA entered the TCA cycle. Particularly in IS, the cycle is downregulated in the first steps: 2-oxoglutarate is replenished via glutaminolysis [29]—a process by which glutamine is converted into glutamate. The activation of glutaminolysis represents an adaptive reaction of cells to produce energy when testosterone is deficient but insulin is active, as in IS but not in IR [29]. Glutamate accumulation in the liver stimulates gluconeogenesis and contributes to the development of glucose intolerance, as described by Newgard et al. [40] for obese subjects. Interestingly, glutaminolysis is upregulated in tumor cells and represents the main source of energy in cancer cells [41], glutaminolysis increasing insulin release. However, glutaminolysis leads to an activation of the malate–aspartate cycle, and in IS patients higher levels of NADH and ATP were recorded. On the contrary, in IR the TCA cycle was strongly reduced, being hindered at the citrate–isocitrate level, which is prevalently used for lipogenesis. Consequently, a reduction in ATP production and increased AMP in the plasma was recorded [28], probably because insulin resistance in muscle cells stimulates the expression of oxidized phosphorylation genes [42].

### 2.2. Lipid Metabolism

It is known that when mammals cannot use carbohydrates to generate ATP, glucose is mostly converted into fatty acids (lipogenesis) for synthesis and storage of TGs in the liver and white adipose tissue [43]. In fact, insulin promotes glucose uptake and regulates triglyceride catabolism through the inhibition of hormone-sensitive lipase [43], indicating that insulin plays a strong role in modulating lipogenesis. Moreover, elevated production of triglycerides in non-adipose tissues, such as the liver, induces the overexpression of lipoprotein lipase and contributes to insulin resistance [43]. Higher free fatty acid concentrations were indeed recorded [42,44], indicating a shift in glycerol consumption toward triglyceride formation. Thus, the livers of males with IR hypogonadism are more prone to lipogenesis, since the higher insulin levels are, the stronger the stimulation of lipogenesis. TG production increases in IR (118.4 mmol/L in IS; 226 mmol/L in IR) [29]. Consequently, in IR obesity increases significantly, as well as BMI, with a value of 30.48 kg/m^2^ recorded with respect to the control value of 22.02 kg/m^2^ [29]. On the contrary, in IS hypogonadism, 3-phosphoglycerol phosphate was not diverted to produce TGs, which decreased, but used prevalently in the glycerol shuttle.

Finally, acyl-carnitine, which is essential for the import of fatty acids into mitochondria, is not produced from acetyl-CoA in IS and IR hypogonadism. Thus, reduced β-oxidation of fatty acids was recorded in both cases, suggesting a possible role of testosterone. β-oxidation of short- and medium-chain fatty acids does not therefore represent an energy source in hypogonadism, while long and branched fatty acids are used. This justifies the increase in fat mass and explains the moderately increased dyslipidemia and increase in body mass index (BMI) observed in hypogonadism. Interestingly, in IR male hypogonadism, more acetyl-CoA is transformed into cholesterol, which increases up to 243 mg/dL [29].

In IR, hypogonadism was associated with a significant increase in sphingomyelin (SM), whereas phosphatidylcholine (PC) was mainly cleaved by activated phospholipase-A2 into Lys-phosphatidylcholine (LPC). In hypogonadal patients, arachidonic acid (AA), also produced through the latter cleavage, was prevalently bio-transformed into leukotriene B4 (LTB4) and not into endoperoxides, from which prostaglandins and thromboxane are derived [45].

### 2.3. Amino Acid Metabolism

In IS hypogonadism subjects, most amino acids do not undergo strong alterations, since amino acids are not employed for energy production. On the contrary, in IR amino acids play a main role, BCAAs (valine and leucine/isoleucine) in particular, which were significantly depleted in plasma and utilized to produce energy through glycolysis and the TCA cycle. BCAAs account for nearly 35% of the essential amino acids in muscle proteins; therefore, their utilization to produce energy in IR causes a decrease in muscle mass. Recent metabolome profiling of obese versus lean humans revealed increased catabolism of BCAAs correlated with insulin resistance [40,46]. In fact, individuals with a lower body mass index but who were considered obese had higher metabolic rates of BCAAs and increased resistance to insulin relative to lean individuals. High BCAA levels in plasma contribute to the development of obesity-associated IR [40,46].

Proline and lysine increase in both IS and IR hypogonadism. These two amino acids participate in collagen fiber formation, and their accumulation in plasma is an indication of slower bone formation and of reduced collagen synthesis. This explains the osteoporosis in hypogonadism [47,48], which is strongly related to testosterone deficiency [49,50], independent of insulin activity.

### 2.4. Other Metabolisms

Degradation of uracil produces β-alanine, the precursor of carnosine. This metabolism is significantly decreased in both IS and IR hypogonadism [28,29], suggesting an effect of testosterone deficiency. Thus, muscle weakness, fatigue and mental confusion increase in hypogonadism. This is in line with Penafiel et al. [51] and Varanoske et al. [52], who reported that high intramuscular carnosine may attenuate fatigue during isokinetic and isometric exercise. In agreement with this, upon orchiectomy, significantly decreased carnosine levels in male mice [51] were easily restored by testosterone replacement. Thus, it is not surprising that β-alanine is a popular supplement used primarily to enhance athletes’ performance, as well as muscle growth, strength and power.

## 3. Metabolomic Comparison between IS and IR Hypogonadism after TRT

Upon the administration of TRT for three months, testosterone levels were restored in both IS and IR hypogonadal patients [53,54], but in IR insulin was slightly reduced (decreasing from 17 to 15 µU/mL), in agreement with Kapoor [23]. Since synergic and/or antagonistic action between testosterone and insulin exists [55,56], it is not surprising that this partial insulin reduction can limit the total restoration of all metabolisms, except for those in muscle and adipose tissue, which were shown to benefit.

### 3.1. Carbohydrate Metabolism

Glycolysis was significantly upregulated upon TRT, better in IS than in IR, indicating improved glucose utilization (Figure 2). In this regard, it has been reported that testosterone increases the expression of GLUT4 in cultured skeletal muscle cells, hepatocytes and adipocytes [39,56], as well as membrane translocation, promoting glucose uptake in adipose and skeletal muscle tissue [57]; therefore, both muscles and adipose tissues benefit from testosterone restoration.

As expected, in IS the pentose phosphate pathway was reduced upon TRT and was paralleled by a decrease in oxidized glutathione (GS-SG). This indicates lower oxidative stress related to testosterone supplementation. In IR, gluconeogenesis (which is the main energy source for hypogonadal IR before TRT) stopped after treatment, confirming the role of testosterone. Biodegradation of branched-chain amino acids stopped, confirming that their increase is correlated with insulin resistance [40,46] and modulated by testosterone.

In both IS and IR, the TCA cycle was not completely used, confirming that in hypogonadism energy is not produced through canonical pathways. In IS, glutaminolysis was stopped and more glutamate was available, blocking the malate–aspartate shuttle. NADH and ATP were restored to control levels. In IR, as a result, although the glycerol shuttle was re-activated upon TRT, total NAD and NADH were significantly lower, as were ATP levels, revealing that, in terms of energy, IS showed a greater benefit than IR (Figure 3).

Interestingly, in both IS and IR, after TRT there were significant increases in lactate and acetyl-CoA production (Figure 4), which were much (ten times) higher in IR [53,54].

Lactate increase represented the first response to testosterone supplementation, in agreement with Enoki et al. [58] and Burns [59]. Lactate and testosterone cause reciprocal effects in Leydig cells [39,60], where lactate stimulates testosterone production and testosterone stimulates lactate production. An increase in lactate was recently associated with type 2 diabetes and insulin resistance [61]. Testosterone deficiency, induced by progressive stages of diabetes mellitus in rodent models, impairs glucose metabolism, favoring metabolic reprogramming toward glycogen synthesis [62].

In both IS and IR, acetyl-CoA increased significantly but was not related to increased b-oxidation of fatty acids, as commonly observed during fasting or when the glucose pathway is hampered, due to several causes. First of all, in both cases, the TCA cycle is strongly reduced. In IR, accumulation of acetyl-CoA exhibits feedback inhibition of lactate dehydrogenase in the absence of insulin. Consequently, the rate of conversion of pyruvate to lactate is decreased and pyruvate is converted into acetyl-CoA by PDH enzymes (see below), commonly downregulated by insulin [63]. Finally, in IR the degradation of leucine/isoleucine also contributes to increase in acetyl-CoA [54], these being two well-known ketogenic amino acids [64,65]. Since leucine/isoleucine and valine account for nearly 35% of the essential amino acids in muscle proteins, after TRT, a higher protein catabolism of skeletal proteins occurred, causing a decrease in muscle mass. In conclusion, in IR, levels of acetyl-CoA are higher and the molecules are completely bio-transformed into ketonic bodies.

### 3.2. Lipid Metabolism

In IS, glycerol 3-phosphate was consumed preferentially with respect to lipid synthesis, reacting with fatty acids and producing more glycerophospholipids and phosphatidylcholine (PC) [45,53]. Lower levels of acetyl-carnitine were still recorded, in agreement with Fukami et al. [66]. Since acetyl-carnitine is fundamental in transporting fatty acids from the cytoplasm to mitochondria for β-oxidation, fatty acids increased and more triglycerides were produced. As a result, lipid accumulation might occur, with a consequent decline in the availability of energy in the heart, skeletal muscles and kidneys [39]. On the contrary, in IR, lower glycerol levels reduced triglyceride formation, increasing the number of fatty acids (FFAs), which were not bio-transformed into ketone bodies by b-oxidation but released into the blood upon TRT. However, in IR, upon TRT, testosterone should stimulate lipolysis in adipose tissue, with increased release of fatty acids, rather than in muscle, with impaired glucose storage. This is the main defect in IR, although some fatty acids are incorporated into triglycerides due to the excess of glycerol. Interestingly, in IR the altered sphingomyelin (SM), phosphatidylcoline (PC) and phospholipase C (LPC) levels were completely restored to control levels [45], and SM, PC and LPC returned to levels similar to those of the controls. In addition, arachidonic acid was newly converted into prostaglandin-A2, thromboxane-A2 and 5(S)-hydroxyeicosatetraenoic acid (HETE), suggesting that testosterone probably plays a role in controlling the hypogonadal alterations reported above. Finally, cholesterol, HDL and lipid metabolism did not show any improvements at 60 days; probably these metabolisms require longer times.

### 3.3. Amino Acid Metabolism

In IS, more histidine was prevalently catabolized into carnosine, which increased, supporting the role of both insulin and testosterone in this metabolism (Figure 3). In IS, skeletal protein catabolism was reduced, and fewer branched-chain amino acids were released into the blood. This agrees with D’Antona [67], who demonstrated the anti-aging role of the BCAAs leucine/isoleucine and valine in mitochondrial biogenesis in mammals, supporting the role of testosterone in the control of muscle protein synthesis [51,68].

In IR, a higher protein catabolism of skeletal proteins was recorded after TRT [54], causing a decrease in muscle mass. A recent metabolome profiling study revealed a correlation between BCAA and insulin resistance [40,46], which remains after TRT. This explains the muscle weakness reported by all patients.

Finally, higher concentrations of proline and lysine were also recorded in both IS and IR, suggesting a positive influence of testosterone on the synthesis of collagen fibers, independently of the presence or absence of insulin. Reduced bone loss and excretion of bone-degradation parameters, such as hydroxyproline, were also recorded [54], in agreement with Tenover [69] and Wang et al. [70].

Figure 5 summarizes the effects of testosterone therapy on glycolytic, lipidic and amino acid metabolisms.

## 4. Discussion

The data reported indicate that, in the case of testosterone deficiency in IS hypogonadism, glycolysis and glutaminolysis produce energy, probably due to the fact insulin increases ATP and NADH levels and promotes glutaminolysis. In IR, instead, the main source of energy is gluconeogenesis, fueled by amino acids, branched-chain amino acids in particular, and the malate–aspartate shuttle [53,54]. However, comparisons revealed that, upon TRT, not all metabolisms were completely restored, and metabolic re-programming was observed in both IS and IR.

The TCA cycle was partially restored in both cases, but ATP levels as well as NADH production were low, indicating that testosterone therapy did not resolve the energy supply through canonical pathways.

Lactate and acetyl-CoA increased in both IS and IR upon TRT. In IS, the reactivation of GLUT4 in muscles induced the activation of an anomalous glucose–lactate cycle, where alanine was excluded from the cycle [53]. This activation included the participation of glucose, which is possible in these patients thanks to insulin still being active. The glucose–lactate cycle becomes the main energy source in highly oxidative cells (e.g., heart, brain and lung cells), and the lactate produced can in turn be converted into glucose in the liver and kidneys for use by muscles.

The higher concentration of acetyl-CoA recorded in IR after treatment with testosterone is due to several causes. It is the result of the balance between the deficiency of acyl-carnitine, essential for the transport of fatty acids from the cytoplasm into the mitochondria (and therefore the b-oxidation of fatty acids cannot occur), and the absence of the inhibitory effect of insulin on their transport into the mitochondria. Probably, in IR hypogonadal patients, b-oxidation takes place but in a reduced rate such that it cannot be the main cause of acetyl-CoA production during fasting or glucose pathway reduction. In both IS and IR, the accumulation of acetyl-CoA recorded is also due to its reduced utilization in the TCA cycle. Moreover, acetyl-CoA, when increased, exhibits feedback inhibition of lactate dehydrogenase in the absence of insulin. Consequently, the rate of conversion of pyruvate to lactate is decreased and pyruvate is converted into acetyl-CoA by PDH enzymes (Figure 6) commonly downregulated by insulin [63]. Moreover, as a consequence of lactate increase, lactate dehydrogenase exhibits feedback inhibition; therefore, the rate of conversion of pyruvate to lactate is decreased, and, for the most part, pyruvate is converted into acetyl CoA by PDH enzymes, which are not downregulated by insulin (Figure 6). Finally, the degradation of leucine/isoleucine also contributes to increased acetyl-CoA [54], these being two well-known ketogenic amino acids [64,65].

In the case of IR, the higher concentrations of acetyl-CoA were preferentially biotransformed into the ketone bodies [54] acetoacetate and 3-hydroxybutyrate (Figure 5). This reaction was catalyzed by 3-hydroxy-3methylglutarylCoa synthase (mtHMGCoA synthase), an enzyme which is inhibited by insulin but overexpressed by testosterone [71]. In agreement, a decrease in 3-hydroxybutyrate was documented upon inhibition of testosterone production in Leydig cells by ethanol [72].

It is of note that ketone bodies were produced only in IR [29] hypogonadism and that the rate of production increased upon TRT, while their production was not observed in IS hypogonadism, neither before nor after TRT [28,53]. It is clear that insulin plays a role in inhibiting ketone body formation. Thus, ketosis seems to be an alternative route for energy supply, having the same metabolic effects as insulin but at the metabolic or primitive control level, bypassing the complex signaling pathway of insulin. Thus, both insulin and ketones seem to produce the same effects on both the metabolites in the first one-third of the TCA cycle and on mitochondrial redox states, increasing the hydraulic efficiency of the well-perfused working heart [73]. It has been reported that the hydraulic efficiency of the heart is 28% greater via the metabolism of ketone bodies compared with a heart that metabolizes glucose alone [74], explaining the adaptative route for energy supply. Unfortunately, after TRT, IR patients showed clinical symptoms related to ketonuria, similar to those shown by individuals following a ketogenic diet, the so-called “keto flu” [75], with psychiatric problems [76]. This must be taken into account before the administration of TRT to IR hypogonadal patients. Recently, circulating ketone bodies have been recorded in individuals with type 2 diabetes [77].

## 5. Conclusions

In conclusion, comparing the metabolisms recorded in cases of IS and IR hypogonadism before testosterone restoration, we can affirm that, independently of the presence or absence of insulin, testosterone deficiency results in lactate decrease, acetyl-CoA decrease, TCA cycle reduction, blockage of the production of acetyl-carnitine (and consequently blockage of the β-oxidation of fatty acids) and blockage of collagen synthesis and carnosine production. In support of this, testosterone therapy, independently of the presence or absence of insulin, induces lactate and acetyl-CoA production, TCA cycle modulation, reduced muscle skeletal protein catabolism, reduced branched-chain amino acid (leucine, isoleucine, and valine) release into the blood, collagen synthesis and carnosine production. Interestingly, upon TRT, acetyl-carnitine metabolism, and consequently β-oxidation of fatty acids, is not activated in the presence or absence of insulin, at least over short time periods.

The reported analysis supports the hypothesis that if hypogonadal patients who still have active insulin (IS) are treated with TRT, IS will not worsen and lead to insulin resistance (IR) such that the metabolisms related to testosterone and insulin cannot be easily recovered. This should prompt endocrinologists to impose testosterone therapy on hypogonadal patients before insulin resistance sets in. Clearly, testosterone deficiency, which can occur in individuals who already have insulin resistance for other reasons than hypogonadism, results in loss of the ability to reactivate all metabolisms.

## 6. Future Directions

This review underlines the importance of using a systems biology approach to elucidate metabolic pathway changes in hypogonadism and for better understanding of the mechanism of “metabolic syndrome” correlated with low levels of testosterone and associated insulin resistance. The new findings will help in selecting patients who will respond to hormone treatment and provide accurate biomarkers for evaluating responses to treatment, eventually leading to better strategies in preventing systemic complications in patients not fit for hormone replacement therapy.

Clinically, testosterone therapy of IR should be integrated with the development of gluconeogenesis precursors as well as supplementation with amino acids, especially leucine, isoleucine and valine. The addition of citrate and other amino acids could help. Carnosine and β-alanine should be supplemented.

The negative effects of the absence of insulin in IR could be better attenuated by administration of metformin.

Carnitine and/or acetyl-carnitine supplementation is recommended for both sub-groups of patients. In support of this, it has been shown that carnitine addition inhibits the development of cardiovascular disease, ameliorates aging-related sexual dysfunction, and reduces levels of free fatty acids [66]. Several studies have emphasized the effect of carnitine as a replacement therapy in the treatment of hypogonadism to improve male re-productive function, making carnitine an appropriate candidate for the therapy of symptoms associated with aging [74,75].

Regarding the ketosis observed in IR after TRT, this could be managed by the utilization of remedies previously proposed for “keto flu” [77]: increasing sodium supplements with electrolytes, drinking broth (including bone broth and stock cubes) and increasing magnesium, potassium and dietary fat intake (including avocadoes, MCT, olives, butter, nuts and fat bombs), as well as increasing water intake.

## Figures and Tables

**Figure 1 ijms-23-12730-f001:**
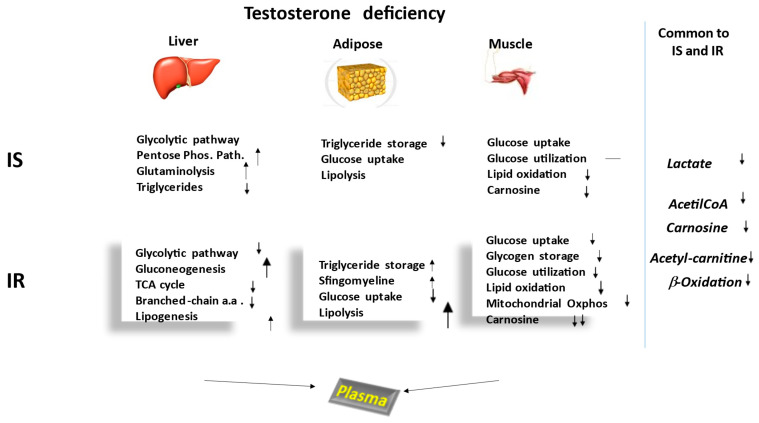
Schematic representation of the metabolisms affected by testosterone deficiency in the three main tissues affected by hypogonadism. Boxes show the metabolisms notoriously altered by testosterone deficiency, the arrows indicating increases or decreases, as well as alterations in terms of intensity. This figure also shows that plasma from a male with hypogonadism is the biofluid in which the metabolites are passively excreted from all other tissues.

**Figure 2 ijms-23-12730-f002:**
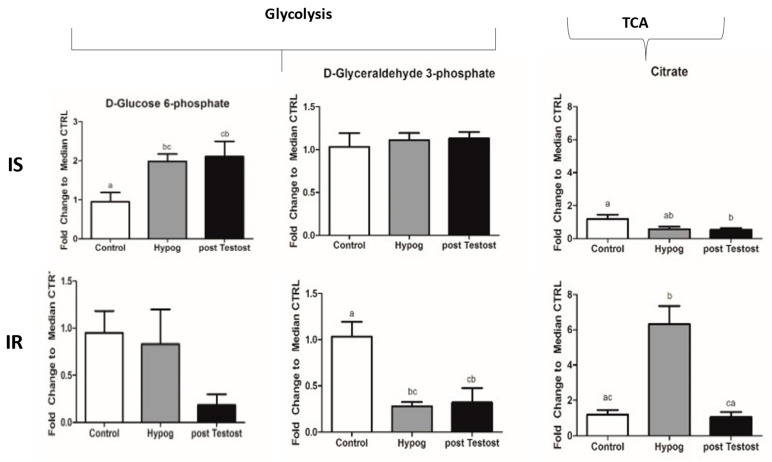
Representative intermediates of glycolysis and the TCA cycle, shown as differences between control subjects and hypogonadal patients, before and after TRT. The data shown are from previous published papers [53,54].

**Figure 3 ijms-23-12730-f003:**
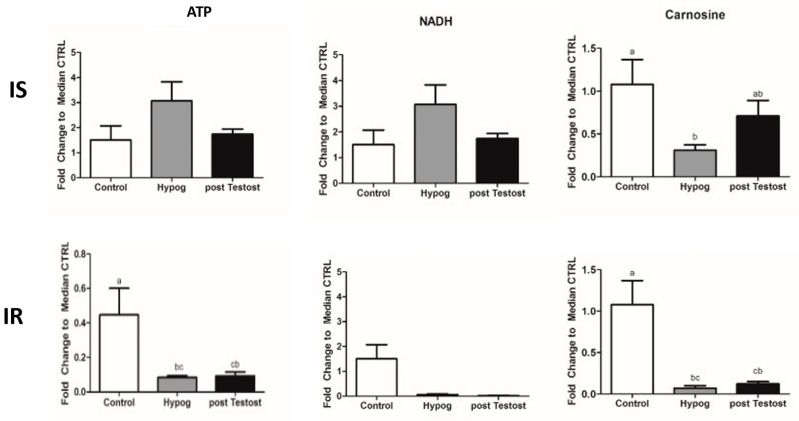
ATP, NADH and carnosine changes in control and hypogonadism subjects, before and after TRT. The data shown are from previous published papers [53,54].

**Figure 4 ijms-23-12730-f004:**
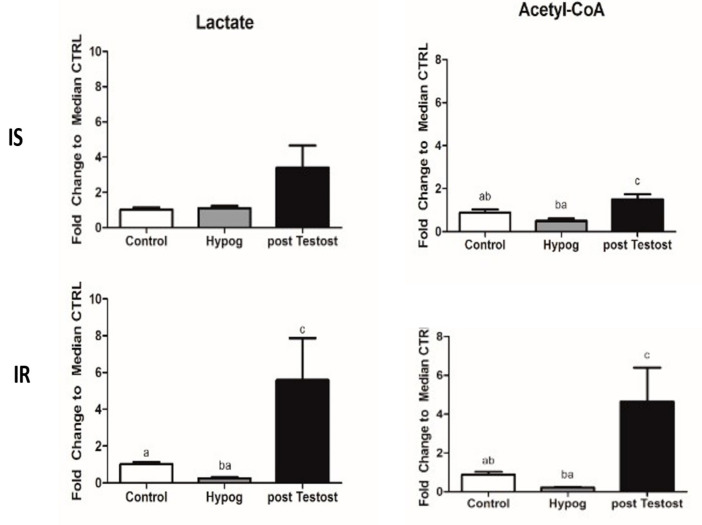
Lactate and acetyl-CoA changes in control and hypogonadism subjects, before and after TRT. The data shown are from previous published papers [53,54].

**Figure 5 ijms-23-12730-f005:**
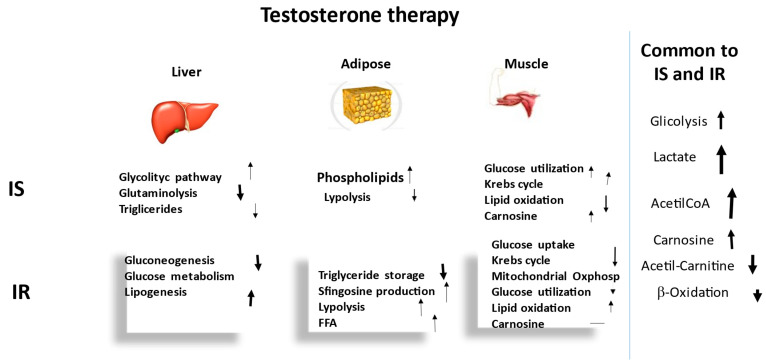
Schematic representation of the metabolisms affected by testosterone therapy in the three main tissues. Boxes report metabolisms notoriously altered by testosterone therapy (increased or decreased). The size of arrow indicates the intensity of the change.

**Figure 6 ijms-23-12730-f006:**
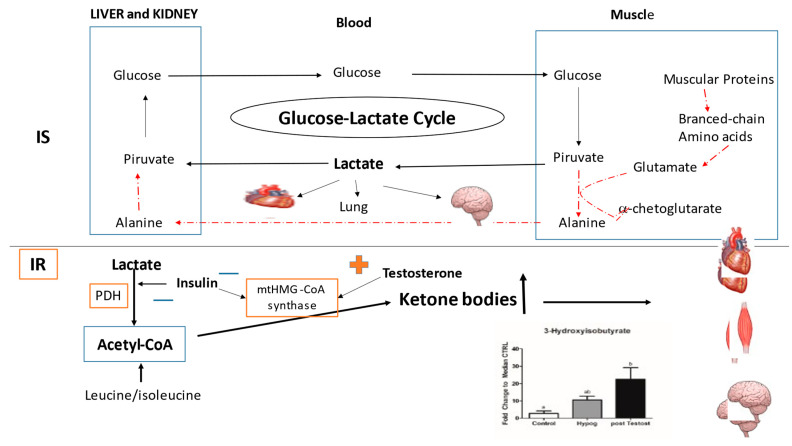
Summary of testosterone therapy effect on most metabolism pathways. The dashed and red lines indicate reduced metabolisms, while the solid lines indicate the positively influenced metabolisms. Arrows indicate increases or decreases.

## Data Availability

Not applicable.

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
