# Peer review of "On the Need to Distinguish between Insulin-Normal and Insulin-Resistant Patients in Testosterone Therapy"

_ijms, 2022, doi:10.3390/ijms232112730_

Round 1

Reviewer 1 Report (Previous Reviewer 1)

The author has made numerous corrections and improvements to the manuscript.  Here I note only a few small corrections yet to be made.

Abstract

Line 12: TCA is not defined.

Lines 16 and 18: Acetyl-CoA should not be capitalized.

Line 20: Possibly “activates” should be “TRT activates”?

Line 23: Testosterone therapy is TRT.

Text

Lines 34 and 35: The author abbreviated testosterone as T in line 34 but then in line 35 again use “testosterone”.  Please be consistent: “T” or “testosterone” but not both.  Personally, I favor “testosterone” over “T”.

Lines 60 and 73: IS is already defined in line 60 so it does not need to be redefined in line 73.

Figure 1: None of the misspellings that I previously pointed out have been corrected.

Line 214: Branched-chain a.a. is not defined and since the author used “branched-chain amino acids“ in 5 other places, the abbreviation here is not needed.

Figure 2: For consistency, it might be helpful to change “Krebs” to “TCA”.

Figure 5: None of the misspellings that I previously pointed out have been corrected. 

Line 364 and 366: Glucose should not be capitalized.

Conclusion

Lines 403 and 409: Acetyl-Carnitine and acetyl-Carnitine should not be capitalized.

Author Response

fig 2 has been split in two to increase its quality and make the message that the data in the figure wanted to convey more incisive. As a consequence, the quality has improved.

Figure 5, now 6, has been redrawn. The quality has improved and now is more understandable.

  The text has been revised and improved, with the help of a native speaker.

The arrows in the figures have been improved and now visible. The different size maintained because it expresses the intensity of the change, previously specified in the text but now in the legends.

Reviewer 2 Report (Previous Reviewer 3)

I recommend acceptance of the manuscript after minor grammatical changes.

Author Response

The grammatical errors and language of the manuscript have been corrected with a native speaker.     

.Plagiarism is now strongly reduced and more bibliography has been added.

Quality of figures improved and Fig. 2 split in two. Figure 5, now 6, has been redrawn

This manuscript is a resubmission of an earlier submission. The following is a list of the peer review reports and author responses from that submission.

Round 1

Reviewer 1 Report

Testosterone therapy need to distinguish the insulin-normal 2 patient from the insulin-resistant one.

This is a fascinating and masterful review of a very complex and interesting field.  I have one major suggestion and a number of minor suggestions.  The overwhelming thought that I had as I read through this thorough review on the interaction of insulin resistance and testosterone therapy was this: Which came first, insulin resistance or hypogonadism?  In conclusion, it was clear that insulin resistance came first and testosterone therapy could only partially reverse the effects.  As only one reader, I would appreciate it if the author would, with a few words, address this larger issue that this paper helps resolve. 

Here a some minor issues to consider.

Abstract

I have made numerous suggestions in the Abstract for two reasons.  First, since it is the most visible part of the paper, it is most important to be exactly correct. Second, as an example of the very careful proofreading that needs to be done on the remainder of the text and figures.  There are numerous obvious language errors that can easily be caught by a careful reading by a second and third colleague.

Line 7: Why is “normo-insulin” abbreviated as IS?  The answer, of course, is that we all know normo-insulin as “insulin sensitive”.  So why not call normo-insulin as insulin sensitive so that it matches its abbreviation?  Conversely, use NS as the abbreviation.

Line 10: Consider changing “into” to “between these”

Line 10: “hygonadism” should be “hypogonadism”.

Line 11: “actives” should be “activates”

Line 11: More modern medical terminology has gotten away from use people’s names.  The Krebs cycle is now more commonly called either the citric acid cycle (CTC) or the tricarboxylic acid cycle (TCA cycle).

Line 12: Consider removing “a process used by cancer cells”.  It only confuses the main issue.

Lines 12, 13: “it is reduced and addressed to lipogenesis.”  What does “it” refer to?  I think it is the Krebs cycle, but it is confusing.  What then is addressed to lipogenesis?  Is it glutaminolysis?  A few more words would help clarify this.

Line 13: “fat” should be “fatty”.

Line 14: In “hampered and FFA increased” I suggest that “hampered by hypogonadism” would be helpful.  Also “FFA” is not defined.

Line 15: “testosterone therapy” is used 3 times in the Abstract but the abbreviation is only given on the second time and never used again.  I suggest giving the abbreviation on the first time and then using only the abbreviation for the second and third usage.

Line 16: The Cori cycle is also called the lactic acid cycle or the glucose-lactate cycle.

Line 16: It is not clear that the sentence beginning “In IS lactate is used….” is still talking about TRT.  I suggest something like this, “After TRT, lactate is used in IS…”

Line 16, 17: “AcetiCoA” is “acetyl-CoA”

1.       Introduction

Line 29: It seems that the “T” abbreviation for testosterone is not needed.  It is used sporadically throughout the text.

Line 33: For “free cholesterol” does the author mean “total cholesterol”?

Line 38: “T” used for testosterone.

2.       Relevant Section

This is a non-descript name for this section. A more accurate title might be “Results” since section 3. is Discussion. An alternative and possible more helpful division would be to turn section 2.1 into 2. and section 2.2 into 3.

Lines 92, 93: More “T”.

Figure 1 is very helpful in understanding how testosterone deficiency affects energy metabolism in the three key organs in IS and IR.  Some suggestions: “Glucose metabolism” is rather vague.  Does the author mean glycolytic pathway? That might be more descriptive and specific. Is it the same as “Glucose utilization”?  What is PPP? No abbreviations explained in the footnotes. “Lypolysis” should be “Lipolysis”. “Triglicerides’ should be “Triglycerides”. “Branced” should be “Branched”. Should “Sfingomieline” be “Sphingomyelin”? “Oxphosp” is usually “Oxphos”. “Acetil-Carnitine” should be “Acetyl-Carnitine”.  Having Carnosine in both of the Muscle groups and also in the Common group is confusing.

Line 111: “TCA” is used as an abbreviation for Krebs cycle.  This suggests that the author might want to start with tricarboxylic acid cycle and use TCA throughout.

Line 130, 131: Note the correct spelling of acetyl-CoA.

In Figure 4 there were many of the same issues as in figure 3.

Author Response

I thank the Referee for the constructive suggestions to improve the text, but above all for suggesting me to finalize the review to give a useful message to endocrinologists. I am a specialist in metabolomic but often miss the clinical significance that the data contain. However, several new sentences have been added in the abstract and Conclusion, as well as corrections have been made in the text.

Although the guide line of journal asks the “relevant data” session I have followed your suggestion “Results”.

Reviewer 2 Report

Zolla L submitted the review manuscript entitled "Testosterone therapy need to distinguish the insulin-normal patient from the insulin-resistant one." While interesting, major editing of the texts and images are necessary:

1. Be consistent with the abbreviation for beta oxidation. Some are B others the word Beta. I suggest to use the greek letter.

2. Some errors in sentence construction. Some sentences form a separate paragraph: Lines 130-131,  173-174, 284-285

3. Major of editing of images is necessary to improve the quality. Fig. 1 and 4 shows arrows with different sizes with some lines not visible. there are arrows that are pixelated.

4. The bar graphs (Fig. 2 & 3) headings and x and y captions are of very poor quality. Changing in font style, size. The labels can't be read in Fig. 2.

5. Wrong spellings: Fosfolipids in Fig. 4

6. The illustration in Fig. 5 looks like copy-pasted. There are spelling mistakes. The images are of poor quality and the schemes very difficult to understand.

7. Using the term glucose metabolism to refer to anabolic processes is vague. The author must be more specific to indicate if glucose catabolic or anabolic processes is affected.

8. Branched aa must be changed to branched-chain amino acids.

Author Response

Quality of figures has been made. The arrows in Figure 1 and 4 have different thicknesses to also indicate the different intensity of the metabolomic change.

Reviewer 3 Report

1.      This is a review article, but the authors presented figures 2 and 3 and a part of figure 5 as a research article. It is not clear from where the data were collected for drawing these figures. If the authors presented the figures from other articles, proper citation should be performed. If the authors performed analysis by summing some articles, that should also be mentioned.

2.      The main problems with the manuscripts are the huge grammatical and typing errors (hygonadism- in abstract, fat acids- in abstract, AcetiCoA- in abstract and other places, Acetil-Carrnitine- inconclusion and other places, B-ossi-dation in abstract and other places,) throughout the manuscript. Therefore, the grammatical errors and language of the manuscript should be corrected with a professional body.

3.      Plagiarism without the bibliography, according to Turnitin, is 53%, which should be reduced.

Author Response

The review is a summary of papers published by me, so even if I tried to change the sentences the concepts are the same as reported in the previous papers. It is a repetition more than a plagiarism. However, I added references.

Quality of figures 2,3 and 5 have been improved as well as text. In the legends of Figure 2 and 3 has been added that data shown are a collection of previous published data. References have been added.

Round 2

Reviewer 2 Report

Although, changes were made to improve the presentation language-wise, i did not see much changes on the figures and paragraph constructions.

Reviewer 3 Report

The authors according with my suggestion. This is not I cannot check the plagiarism due to the lack of a clean copy of the revised version. If plagiarism or similarity is within the acceptable range, the manuscript may be accepted.